# Experimental Investigation of the Effect of Groundwater on the Relative Permeability of Coal Bodies around Gas Extraction Boreholes

**DOI:** 10.3390/ijerph192013609

**Published:** 2022-10-20

**Authors:** Mingkun Pang, Hongyu Pan, Hang Zhang, Tianjun Zhang

**Affiliations:** 1College of Energy, Xi’an University of Science and Technology, Xi’an 710054, China; 2College of Safety Science and Engineering, Xi’an University of Science and Technology, Xi’an 710054, China; 3Key Laboratory of Western Mine Exploitation and Hazard Prevention of the Ministry of Education, Xi’an 710054, China

**Keywords:** coal gas control, gas pre-extraction, gas extraction borehole, permeability of coal body, effect of groundwater

## Abstract

Water infiltration in boreholes is a common problem in mine gas pre-extraction, where water infiltration can significantly reduce the efficiency of gas extraction and curtail the life cycle of the borehole. It is important to evaluate the effect of groundwater on the permeability of the coal body around a gas extraction borehole. In order to determine the seepage parameters of the fractured coal body system around the borehole, a water–gas two-phase seepage test was designed to determine the relative seepage parameters of the fractured coal media seepage system. The main conclusion is that the relative permeability of gas can be effectively increased by increasing the negative extraction pressure at the early stage of extraction to accelerate drainage to reduce the water saturation of the coal seam. Under the combined effect of porosity and seepage pressure, the relative permeability of gas and water in the fractured coal rock body shows three stages. The dependence of the total permeability on the effective stress is closely related to the stages in the evolution of the pore structure, and the total effective permeability decreases with the increase in the effective stress. A decrease in porosity can lead to a decrease in permeability and an increase in the non-Darcy factor. Through an in-depth analysis of the damage and permeability pattern of the coal body around the perimeter of the dipping borehole, the efficient and safe extraction of gas from dipping boreholes in water-rich mines is thus ensured.

## 1. Introduction

The efficient pre-extraction and extraction of coal seam gas resources can not only reduce gas disasters, but also play an essential role in optimising China’s energy structure and contributing to “carbon neutrality” [1,2]. At present, requirements of “four” extraction management steps (extraction maximisation, extraction standardisation, extraction refinement, and extraction informatization) should be met [3]. The amount of gas extracted from the surface is only 0.45 × 10^10^ m^3^, while the amount of gas extracted from underground boreholes reaches 1.23 × 10^10^ m^3^, accounting for 73.2% of the total amount of gas extracted [4]. In order to improve the extraction efficiency of gas energy, borehole gas pre-pumping has become an important means of gas management in high-gas mines, and gas control in high-gas mines is also an important part of ensuring safe production in coal mines [5]. However, as the mining depth increases, the hydrogeological conditions of the mine become more and more complex. The presence of overburden aquifers can pose a risk of borehole flooding for coal seam gas pre-pumping and pose a huge challenge for underground gas control [6,7]. Following the construction of the borehole, negative pressure is generally applied to improve the efficiency of gas pre-pumping [8]. Although, this method has been widely adopted in China, there are still some issues that need to be addressed.

In order to improve the efficiency of gas extraction, several researchers have conducted theoretical analyses and achieved some meaningful results [9]. However, the gas pre-pumping efficiency of applying water dipping to boreholes is mainly governed by several factors. Firstly, water immersion causes softening and expansion of the coal body around the hole, which substantially reduces the stability of the borehole [10]. The reason for this is that water has softening, sludging, and erosive effects on the coal body, and the infiltration of seam water accelerates the process of creep deformation and crack expansion of the perimeter coal body in the borehole, which is reflected in the expansion and swelling of the perimeter coal body in the borehole on a macroscopic scale [11]. Secondly, the seam water will occupy the perimeter coal gas infiltration channels, making it difficult for the perimeter gas to flow into the borehole. This is evinced by the fact that seam water infiltrating into the borehole will occupy the pore/fissure space of the perimeter coal body, partially or completely filling the pores/fissures within the coal body, preventing the necessary passage for gas pre-pumping around the borehole [12]. Finally, the resistance to water–gas penetration in both phases of the perimeter coal body is greater, which increases the need for negative extraction pressure. The essence of this is that the borehole is permanently submerged in a waterlogged geological environment and the presence of water increases the difficulty of gas flow through the permeable channels of the coal body, i.e., the gas becomes “trapped” in the perimeter coal body during extraction [13].

Therefore, it is necessary to reveal the softening mechanism of water on a coal body, the formation process of seepage channels and the establishment of a coupled seepage model. Based on a three-dimensional spatial fractal model of particle size distribution, Tyler [14] investigated the fractal characteristics of particles after creep damage of perforated coal rocks and characterised the fractal dimension by using a formula based on particle mass. Ouyang et al. [15] and Zhang et al. [16] proposed a fractal analysis method based on a nuclear magnetic resonance technique to visualise the fractal characteristics of pore rupture in coal bodies.

By observing the fracture morphology within the fracture zone of the borehole, Zhou et al. [17] found that there were mainly three morphologies—straight fractures, bent fractures, and bifurcated fractures—within the fracture zone of the borehole. In the process of studying the percolation of single fractured rock samples, Hu et al. [18] derived the formula for calculating the permeability coefficient by establishing a physical model under three-dimensional stress. He concluded that the lateral stress on fractured rock samples follows a negative exponential trend in terms of their permeability coefficients. From the geometric roughness of fracture walls, Wang et al. [19] (among others) established an equivalent calculation method for parallel plates with discrete units. The relationship between the gap width and flow rate of rough fissures was given. Considering that the pore and fracture spaces of the perimeter coal body will be filled with water following water immersion into the borehole, Pusch et al. [20] suggested that the difference in concentration causes the gas to dissolve and diffuse into the pore water of the coal body firstly, and then enter the borehole under the action of negative extraction pressure. From the perspective of flow field distribution in porous media, Vatani et al. [21] proposed that a diffusion field and a seepage field existed simultaneously within the peripore coal and considered diffusion and seepage as a mass exchange process. In the study of the subsurface water–gas two-phase seepage process, Zhang et al. [22] combined the Buckley–Leverett equation, the continuity equation and Darcy’s law to develop a three-dimensional dynamic coupling model of water–gas two-phase seepage. Based on the condition that gas is dissolved in water, Ceccato et al. [23] developed a multiphase fluid–fluid–solid coupling model under the combined influence of multiple parameters of coal body deformation field and seepage field during coalbed methane mining.

Therefore, the extraction process of water infiltration boreholes has certain peculiarity in terms of borehole deformation, fracture evolution, and gas infiltration. On the one hand, the infiltration of water affects the fracture development and damage pattern of the coal body around the hole. On the other hand, the presence of water in the coal body can significantly reduce the efficiency of gas pre-pumping and decrease the life cycle of the borehole. The deformation of the ‘ring’ of fractured coal media formed around the borehole is often subject to a combination of external and internal stresses. Considering the specificity and importance of the deformation of this fractured coal media, this study explores the main factors affecting the deformation of the pore structure in order to obtain accurate values of the water–gas relative permeability. The decisive influence of the pore structure on the seepage system of the fractured coal media is evaluated, which is an important means of characterising the compaction deformation process in porous media. By analysing the permeability characteristics of the coal body around the dip borehole, the efficient and safe extraction of gas from dip boreholes in water-rich mines is ensured.

## 2. Theory

For gas extraction boreholes, long-term water infiltration can lead to blockage of the perimeter coal gas infiltration channel, which severely inhibits the diffusion and convergence process of gas within the perimeter coal and reduces the efficiency of gas extraction from the borehole [24,25]. The fracture structure around the borehole is unique in that it is dense, with small displacements between particles and many fine pores. These pore and fracture networks are the main pathways for gas flow in the coal body [26]. Therefore, the study of the permeability parameters of the fractured coal body around the borehole is extremely important, mainly involving the calculation of the pore parameters of the fractured coal body and the calculation method of the relative permeability of the fractured coal body.

### 2.1. Calculation of Pore Parameters of Fractured Coal Media

For a special seepage system consisting of a fractured coal body, when the seepage reaches a steady state, the pore pressure decreases linearly with the seepage inlet as the origin and the direction of seepage as the positive direction of the *x* axis [27,28], and then the pore pressure values *p*(*x*) at each point can be obtained as:(1)$$p(x)=p_{1}-\left(\frac{p_{1}-p_{2}}{H_{s}}\right)$$
where: *p*_1_ is the pore pressure at the inlet end of the seepage, MPa; *p*_2_ is the pore pressure at the outlet end of the seepage, MPa.

In the experiments described in this paper, since the seepage outlet is connected to the atmosphere, the pressure at the outlet end of the seepage is taken to be 0, i.e., *p*_2_ = 0. The pore pressure gradient can then be expressed as:(2)$$G_{p}=\frac{p_{2}-p_{1}}{H_{s}}=-\frac{p_{1}}{H_{s}}$$
where: *G*_p_ is the pore pressure gradient, MPa/m. The effect of formation water on the permeability of the coal body around the perimeter of the gas extraction borehole is shown schematically in Figure 1.

### 2.2. Calculation of the Relative Permeability of Fractured Coal Media

The relative water–gas seepage parameters in fractured coal bodies can be calculated according to Chinese National Standard GB/T 28912-2012, where the water saturation can be deduced from the weight difference in samples at different times.
(3)$$S_{w}=\frac{m_{i}-m_{0}}{V_{p}\rho_{w}}\times 100\%$$
where: $S_{w}$ is the water saturation of the sample; $m_{i}$ denotes the total mass when containing water, g; $m_{0}$ is the total mass when dry, g; $V_{p}$ is the pore volume of the sample, m^3^.

During the test, each phase of the fluid satisfies the generalised form of Darcy’s law, so that the effective permeability of each phase of the fluid can be expressed as:(4)$$\left\{\begin{array}{l} K_{\mathrm{eg}}=\frac{2p_{a}Q_{g}\mu_{g}H_{s}}{A\left(p^{2}-p_{a}{}^{2}\right)}\times 10^{2}\\ K_{\mathrm{ew}}=\frac{Q_{w}\mu_{w}H_{s}}{A\left(p-p_{1}\right)}\times 10^{2} \end{array}\right.$$
where: $K_{\mathrm{eg}}$ is the effective permeability of the gas, mD; $p_{a}$ is 0.1 MPa; μg is the coefficient of viscosity of the gas, mPa-s; A represents the cross-sectional area of the sample, cm^2^; p is the pressure at the inlet end, MPa; $K_{\mathrm{ew}}$ refers to the effective permeability of the water; $\mu_{w}$ is the coefficient of viscosity of the water, mPa-s; $p_{1}$ is the pressure at the outlet end, MPa.

In the relative permeability of water and gas, the absolute permeability can be expressed as the permeability of the gas in the bound water state, and then the formula for calculating the relative permeability can be expressed as:(5)$$\left\{\begin{array}{l} K_{\mathrm{rg}}=\frac{K_{\mathrm{eg}}}{K_{g}\left(S_{\mathrm{ws}}\right)}\\ K_{\mathrm{rw}}=\frac{K_{\mathrm{ew}}}{K_{g}\left(S_{\mathrm{ws}}\right)} \end{array}\right.$$
where: *K*_rw_ is the relative permeability of water; *K*_rg_ refers to the relative permeability of gas; *K*_g_(*S*_ws_) is the permeability of gas in the bound water state.

Considering the softening and erosive effect of water on the coal body when water is immersed in the borehole, the borehole walls will be damaged to varying degrees. Meanwhile, the pore and fissure spaces of the coal body around the hole will be filled with water, the hole is generally wet, and some will show traces of overwatering. The concentration difference allows gas to first dissolve, then diffuse into the pore water of the coal body and then enter the borehole under the negative pressure of extraction. A realistic view of the borehole peephole deformation and damage after groundwater erosion is shown in Figure 2.

## 3. Materials and Methods

The effect of the permeability of fractured coal grains around a hydrometric gas extraction borehole was experimentally investigated. Samples were taken on site and obtained by controlling the Talbol index *n* to samples with different initial porosities. Test conditions were set to control the stress environment, and tests were carried out using a self-designed water–gas two-phase permeameter. The steady-state permeability method was used to measure the permeability base parameters of the fractured coal media.

### 3.1. Materials

#### 3.1.1. Research Background

The sample was selected from the Heyang coal mine in Shaanxi Province, which is in the eastern part of the Weibei coalfield, and the commercial coal is a scarce high-quality power coal in China. He-Yang has a design production capacity of 1.2 Mt/, and this study occurred during mining of the 5# coal seam. The mining method was originally designed for integrated mechanised top-loading coal mining. The 5# coal seam has an original moisture content of 0.76% and is not of good size. The 5# coal seam is a typical “three soft” coal seam, with an unstable roof, a likelihood of falling, and a loose seam with a solidity factor such that f ≮ 1.0; it is easily flaked and has a soft base plate with low compressive strength, making it liable to bottom drumming. Soft coal seam mining produces a lot of dust, and it is difficult to seal holes when injecting water into soft coal seams. Most of the soft coal seams have undergone metamorphism to some extent. The 5# coal seam consists of grey-black siltstone and sandy mudstone (containing 4# coal) at the direct top, grey charcoal mudstone at the pseudo-top and grey fine-grained sandstone at the old top. The thickness of the 5# coal seam varies considerably, reaching 7 m at its thickest, and the average thickness of the coal seam is 4.5 m. The direct bottom is a sandy mudstone, 0.3 m thick and dark grey, with no laminae developed and an agglomerate structure, containing many plant fossils, that swells upon wetting. The old bottom is K3 quartz sandstone or siltstone, 5.0 m thick, grey or dark grey, dense and hard, containing mainly quartz with mica fragments, in which fissures are more developed. The 5# coal seam is more commonly filled with water; it contains a K middle sandstone aquifer at the bottom of the Lower Stone Box Formation, K4 sandstone aquifer at the bottom of the Shanxi Formation, K2 tuff aquifer of the Taiyuan Formation, and an O-ash aquifer.

#### 3.1.2. Samples Preparation

The test coal samples were selected from the in situ coal rock of the Heyang coal mine in Shaanxi Province. The hydrogeological conditions of this mine are complex and water accumulation in the extraction pipeline is common and significant in volume during gas extraction, impairing the efficiency of gas extraction. To study the transport pattern of gas in the fractured coal body and the deformation and damage characteristics of the fractured coal body under different water contents, the in situ coal body was obtained from the mine and transported to the laboratory for indoor research. The fractured coal rock body is not composed of a single grain size coal rock body particle, but by multiple scales of coal particles. In the present work, intact coal samples taken from the site were fractured into small grain size coal samples using a hammer crusher and then the fractured coal samples were screened using a vibrating sieving screen.

To reduce the influence of particle size effect on the test results, four basic particle size intervals of 0 to 5 mm, 5 to 10 mm, 10 to 15 mm, and 15 to 20 mm were selected for the preparation of the fractured coal mixture. To overcome the dimensional catastrophe caused by the mixture of multiple particle sizes and to control the initial grading and initial pore structure of the same group of samples in a consistent manner, the percentage and mass of coal particles in each particle size interval of samples with different initial pore structures were calculated according to the Talbol continuous grading equation [29,30].
(6)$$P_{i}=\left\{\begin{array}{ll} \left[{\left(\frac{d_{i}}{D}\right)}^{n}-{\left(\frac{d_{i-1}}{D}\right)}^{n}\right]\times 100\% & ,\left(i>1\right)\\ {\left(\frac{d_{i}}{D}\right)}^{n}\times 100\% & ,\left(i=1\right) \end{array}\right.$$
where: $P_{i}$ is the proportion of coal samples in the *i*th size range in the sample, *i* is 1, 2, 3, 4; *d_i_* is the upper size limit of the *i*th size range, mm; *D* is the maximum size in the sample, mm; *n* is the Talbol power index.

According to the Talbol continuous grading equation [31], the initial pore structure of the samples differed as the Talbol power index increased and the proportion of coal samples with large grain size intervals in the samples increased. Therefore, to control the effect arising from the initial structure, the Talbol power index was taken as 0.2, 0.4, 0.6, and 0.8 by using an aliquoting method to control the initial structure of the same group of samples to remain consistent. The total mass of each group of coal samples was set to 500 g according to the volume of the permeameter cylinder.

A corresponding mass of fractured coal particles is weighed and mixed by means of a high-precision electronic balance. To avoid manual errors in the initial structure, the samples need to be mixed thoroughly to make them homogeneous. The percentage of particles and the preparation process for the samples are shown in Figure 3.

### 3.2. Methodologies

#### 3.2.1. Experiment Design

The gas and water were selected for isobaric mixing, i.e., the seepage pressure of gas and water was always set to the same value, based on the force in the fracture zone around the hole and the maximum value of the coal seam gas pressure. The principle of testing the water–gas two-phase seepage parameters of the fractured coal body is shown in Figure 4.

During the test, the axial and seepage pressures were taken as follows: the test was divided into five groups of ZS-01, ZS-02, ZS-03, ZS-04, ZS-05 according to the axial pressure gradient, and the axial load was applied at a constant loading rate of 0.05 kN/s for five axial pressures (5 kN, 10 kN, 15 kN, 20 kN, and 25 kN), respectively.

#### 3.2.2. Experimental Equipment

The data acquisition device includes a paperless recorder, computer, and various sensors. The main sensors include flow transmitters and pressure transmitters, among which water and gas flow transmitters are divided into the corresponding turbine-type and vortex-type transmitters. The vortex-type flow transmitter measurement range is 1 to 20 m^3^/h, the turbine-type flow transmitter measurement range is 0 to 1 m^3^/h, using a flow transmitter pressure resistance of 6.3 MPa, and the pressure transmitter measurement range is 0 to 10 MPa. The measuring range is 0 to 10 MPa. The accuracy of the above sensors is 0.5%, and the power supply is 220 V (AC).

The flow, pressure, and mass transmitters in the test pipeline are connected to a paperless recorder, of which the paperless recorder model GT71R0612S2T6V0, powered by AC at 100 V to 240 V, has eight acquisition channels, 64 MB of memory, and a sampling frequency of 1 Hz. The logger is connected to the computer via a 485 USB adapter for the collection of test data; these are used in conjunction with the host computer to visualise the data conversion and segmented free selection. At the same time, the validity of the test and the accuracy of the data can be analysed by computer against the control parameters in the water and gas supply unit. The system is connected by the means of a seepage pipe with a pressure rating of not less than 10 MPa. The flow of fluid is unidirectional, and therefore a one-way valve is required in the inlet and outlet pipes of the seepage meter to ensure the unidirectional flow of fluid.

According to the design and selection of the above components, a DDL600 electronic universal testing machine, two-phase permeameter, water and gas separation device, high-pressure water pump, high-pressure gas cylinder, and pressure-reducing valve were used. A pressure transmitter, flow transmitter, electronic balance, computer, etc. comprised the equipment for operation and monitoring of auxiliary items, with a sorting sieve, electronic balance, felt, steel ruler, stopwatch, etc. also being used. The system is a gas–liquid-coupled seepage test system for fractured coal bodies. The test system can not only study the compaction and seepage characteristics of water–gas in fractured coal bodies. Rather, it can also complete the seepage tests of other crushing media, realise seepage tests of gas and water single-phase and water–gas two phase under the control of different pressure gradients within the measuring range and attain the high-precision global data acquisition of seepage parameters. The water–gas two-phase seepage test system for fractured coal bodies is shown in Figure 5.

#### 3.2.3. Experimental Procedure

The water–gas relative percolation test of fractured coal bodies was carried out via the steady-state percolation method. In this vein, axial pressure was used to control the pressure-bearing environment, and the relative percolation characteristics of the samples under different percolation pressures and different axial pressures were determined by adjusting the magnitude of the percolation pressure. The procedure of the water–gas relative seepage test is based on the operation of the water–gas two-phase seepage test system for fractured coal bodies:Loading the well-mixed sample into the penetrometer, a pre-load of 0.1 kN was applied to the sample, and the initial height H was recorded.The press was turned on to apply a predetermined load of 5 kN to the sample and this load was kept unchanged.In the constant-pressure phase, the seepage pressure of water and gas was adjusted to 0.2 MPa, respectively, by means of a seepage pressure control system. This was maintained for more than 15 s at steady state, and parameters such as flow rate and pressure were collected.The fluid inlet and outlet channels of the permeameter were closed, the weight of the permeameter and sample were weighed and recorded after the pressure had been removed; then, the permeameter was recovered and the press was controlled for reset.The seepage pressure was continuously adjusted to the next stage, and this step was repeated until the scheduled seepage pressure test under that pressure stage was completed.After completing the two-phase seepage test at this pressure stage, the two-phase seepage apparatus was cleaned, the water and gas separation device was drained, and the sample was reloaded to complete steps (1) to (6).

Here, helium was used instead of gas and the viscosity coefficient of helium was 1.89 × 10^−5^ Pa·s. The parameters of water were taken as those used in testing, with a viscosity of 1.5188 × 10^−3^ Pa·s at a temperature of 5 °C and a density of 1 g/mL. The test was conducted three times with the same parameters, and the results were averaged to reduce randomness.

## 4. Results and Discussion

To study the permeability flow pattern of fractured coal grains around the perimeter of extraction boreholes, a water–gas two-phase permeability test platform for fractured coal bodies was developed independently. This platform can be used to estimate the influences of the pore structure on the permeability parameters of the fractured coal media, and thus to study the changes in relative permeability under different permeability pressures. The effect of effective stress on the pore permeability system can then be investigated, and the evolution of the pore characteristics parameters of the permeability system of the fractured coal body can then be studied.

### 4.1. Effect of Porosity on Relative Permeability

Water saturation is a key factor affecting the relative permeability characteristics of water–gas. The distribution of gas and water in the pore space will change as the water saturation changes, and the distribution of water content saturation is mainly directly affected by factors such as porosity and pore pressure [32,33].

Permeability is a characterisation of the ability of the coal body to allow the passage of fluids. To analyse the effect of porosity on the permeability of the two phases with water and gas, the relationship between porosity and effective permeability was plotted according to the test results (Figure 6), taking the effect of seepage pressure on the permeability of the two phases from 0.2 to 0.8 MPa as an example.

As can be seen from Figure 4 and Figure 5, when the seepage pressure is known, the effective permeability to water increases with increasing porosity, an effect which is caused by the increase in the volume content of water in the sample with the increase in porosity. The gas effective permeability of the sample also increases with the increase in porosity, but the increase value gradually decreases with the increase in porosity. When the porosity is less than 0.4872, the gas effective permeability tends to be stable. When the seepage pressure is 0.6 Mpa and 0.8 Mpa, the quadratic term coefficient of water effective permeability is greater than 0, showing that the rate of increase in water effective permeability increases gradually with the increase in porosity. The results of the tests at different permeability pressures imply that the overall effective permeability of water and gas in the samples increases with the increase in porosity, which is consistent with the positive correlation.

Due to the competing percolation between water and gas, the samples differ in their ability to circulate the two phases of fluid as the effective permeability increases. So again, using a seepage pressure of 0.2 to 0.8 Mpa as an example, the relationship between porosity and the relative permeability of the two phases is given (Figure 7).

As can be seen from Figure 7, the relative permeability of water shows the same change pattern as the effective permeability when the permeability is near 0.2 Mpa. However, when the permeability is greater than 0.2 Mpa, the quadratic term coefficient of the fitting function of water relative permeability and porosity are greater than 0. The relative permeability of water shows a trend of decreasing first, then increasing, distinguishing the monotonic increment of the effective permeability. This is because with the change in porosity, the absolute permeability of the sample also changes. This is due to the asynchronous nature of the absolute permeability and water effective permeability change amplitude, resulting in first the decrease and then increase in the water relative permeability. The quadratic term coefficients of the gas relative permeability and porosity fitting functions are also greater than 0, showing the opposite trend to the effective permeability. This is because, as the porosity increases, the saturation of the sample increases, and the competitive permeability relationship between water and gas results in the effective permeability of the sample gas increasing at a lower rate than the absolute permeability of the sample.

Meanwhile, the relationship between porosity and effective permeability and relative permeability all satisfy the quadratic function and have a high degree of fit. The resulting regression equation is:(7)$$K_{\mathrm{er}}=a+b\varphi +c\varphi^{2}$$
where: *K*_er_ is a general term for effective and relative permeability; *a*, *b*, and *c* are fitting coefficients related to the porosity.

### 4.2. Effect of Permeability on Relative Seepage

The change in seepage pressure not only affects the distribution of water saturation, but also directly affects the flow pattern of water and gas. When the axial pressure is known, i.e., the porosity of the sample remains constant, the absolute permeability of the sample will remain constant, and the effective permeability of the sample and the relative permeability show a synchronous change, thus applying pressure to the relationship between relative permeability and seepage (Figure 8).

From Figure 8, the total relative permeability is less than 1 during the test, and with the increase in seepage pressure, the total relative permeability of the sample decreases rapidly and tends to stabilise. On the left side of the isotonic point, the gas relative permeability decreases rapidly; on the right side of the isotonic point, the gas relative permeability decreases slowly and tends toward a stable value. This shows that there is a critical value (phase) of the gas relative permeability influenced by the seepage pressure, while the water relative permeability increases steadily with the increase in seepage pressure and does not show a particular phase. The relationship between permeability and seepage pressure for different axial pressures also follows the same pattern, which is consistent with the variation in water saturation of the sample with seepage pressure. The relationship between gas permeability and seepage pressure is a power function with a goodness of fit greater than 0.98, and the relative permeability of water and seepage pressure are linearly related with a goodness of fit exceeding 0.96. Fitting the seepage pressure and relative permeability gives a better fit, as follows:(8)$$\left\{\begin{array}{l} K_{\mathrm{rg}}=mP^{n}\\ K_{\mathrm{rw}}=M+NP \end{array}\right.$$
where: *M*, *N*, *m*, and *n* are the fitting coefficients related to the seepage pressure. The difference between the effective permeability fitting function for gas and water and the relative permeability fitting function is a multiple of the absolute permeability $K_{g}\left(S_{\mathrm{ws}}\right)$.

These results show that there is a phased pattern of change in water content saturation and gas permeability, with the flow rate and water content saturation increasing when the seepage pressure increases and the effective permeability and relative permeability decreases. A critical value for the influence of seepage pressure can be determined in conjunction with the variation characteristics of gas flow. To this end, the variation pattern of gas flow rate with seepage pressure at different axial pressures is given.

The gas flow rate increases with the increase in seepage pressure, but the amplitude of the increase gradually decreases. Combined with the results of five groups of tests, the change in gas flow rate in the process of water–gas two−phase seepage of the fractured coal body will appear to be at its peak; the peak flow rate area in this paper ranges between 1300 mL/s and 1350 mL/s, that is, the peak line of flow rate is 1300 mL/s. When the gas flow rate enters the peak area, the gas flow rate increases rapidly. The intersection of the peak line and the gas flow rate variation curve is between 0.4 and 0.6 Mpa. The effect of seepage pressure on the gas flow rate of the coal crushing media seepage system is illustrated in Figure 9.

The results of the water–gas relative permeability test of the fractured coal rock body under different pressures indicate that the water and gas permeability in the fractured coal rock body show three stages under the influence of permeability pressure. These are gas permeability, isotonicity, and water permeability. In summary, the critical seepage pressure of gas permeability is determined to be 0.4 to 0.6 Mpa, and when the seepage pressure is greater than the critical seepage pressure, the gas flow rate, effective permeability, and relative permeability all tend to be stable.

### 4.3. Characterisation of the Evolution of the Relative Permeability Curve

The relative permeability curve is the key to the study of water–gas relative permeability. The changing characteristics of the curve can not only reflect the changing pattern of relative permeability, but also the ease of gas flow [30,31]. The relationship between relative permeability and water content saturation is displayed in Figure 10.

As seen from Figure 10, the curvature of the gas relative permeability curve is more concave than that of the water relative permeability curve. This shows that the gas is more sensitive to water saturation, and that its the permeability changes are more pronounced. This is because, as the water saturation increases, the pore volume occupied by the gas decreases, i.e., the gas content of the fluid mixture decreases, the flow pattern of the fluid will change from elastomeric to vesicular flow, and the apparent viscosity coefficient of the gas increases as the pressure increases. The results of the five levels of axial pressure tests imply that, as the water saturation increases, the relative permeability of the gas decreases, rapidly followed by a slow decrease. The average increase in water saturation in the five groups of samples is 22.4%. During the increase in water saturation, the average decrease in gas relative permeability is 67.59% and the average increase in water relative permeability is 19.69%. The increase in gas relative permeability is much larger than the decrease in water relative permeability, indicating that the water saturation exerts a greater influence on gas transport.

With the change in axial pressure, the water saturation of the initial stage of the sample gradually decreases, increasing the initial stage of the relative permeability of the gas, shifting the isotonic point to the left, decreasing the critical water saturation ratio, while the critical relative permeability remains quasi-constant. This is because the seepage pressure directly affects the size of the water content saturation, but the sample water–gas two-phase fluid in the isotonic point of the permeability of the difference is small and changes the isotonic point as a result of the effects of water saturation. At the same time, the water saturation of the sample decreases under the action of axial pressure, and the initial water saturation of the sample decreases from 69% to 60% as 5 kN increases to 25 kN, while the variation in water saturation with pressure at 1.2 Mpa decreases by only 3%, increasing the range of variation in water saturation with the increase in axial pressure. This is mainly due to the inconsistency in the distribution and permeability of water and gas in the sample which is influenced by the porosity and seepage pressure. Therefore, the coupled permeability characteristics of peripore water–gas cannot be described by the relationship between fluid saturation and permeability alone, but the influences of porosity and permeability pressure on permeability should also be considered.

### 4.4. Determination of Relative Coal Crushing Rate in Relation to Seepage Parameters

The re−breaking and transport of the sample will affect the permeability and tortuosity of the seepage channel, thus affecting the seepage characteristics of the coal body. The relationship between the relative coal crushing rate, and the permeability and non−Darcian flow factor is given as an example of the permeability characteristics of the sample at 30 kN (Figure 11 and Figure 12).

As seen from Figure 11 and Figure 12, the permeability of the sample decreases with the increase in the relative coal crushing rate. This is because, as the relative coal crushing rate increases, the proportion of coal bodies in the small particle size range increases, the pore structure of the sample gradually becomes denser, and the tortuosity of the percolation channels increases. This increases the resistance to gas transport and decreases the permeability of the coal body.

The influences of water content and relative coal crushing rate on permeability are dominated by relative coal crushing rate. The source of the difference in permeability at different water contents is mainly due to the softening effect of water on the coal body, resulting in an increase in relative coal crushing rate. The effect of relative coal crushing rate on permeability can be expressed as a linear function, with a goodness of fit exceeding 0.95. There is a negative linear relationship between relative coal crushing rate and porosity, and a cubic relationship between porosity and permeability, but a linear relationship between relative coal crushing rate and permeability. This indicates that the change in porosity is not only due to the fragmentation and rupture of the coal body, but also the involvement of transport deformation of the coal body. A comparison of the relative coal crushing rate and the parameters fitted to the permeability parameters is provided in Table 1.

The relationship between the relative coal crushing rate and the non-Darcian flow factor in Figure 12 shows that the non-Darcian flow factor of the fractured coal body increases with the increase in the relative coal crushing rate, while the non-Darcian flow factor of the sample decreases with the increase in the water content. When the relative coal crushing rate is lower than 0.18, the increase in non-Darcian flow factor is relatively slight, with an average increase in 2.5107 m^−1^. Conservely, when the relative coal crushing rate is greater than 0.18, the increase in non-Darcian flow factor is relatively rapid, with an average increase in 5.67107 m^−1^. This latter value is twice as much as that of the former, and the slope is much greater than that of the former also. This indicates that the non-Darcian flow characteristics of the fractured coal body are exacerbated by the relative coal crushing rate of the sample when the relative coal crushing rate reaches a certain value, i.e., the proportion of coal particles in the small and body size range of the sample reaches a certain value.

### 4.5. The Effect of Effective Stress on the Pore Permeability System

Effective stress is the most basic parameter used to describe the mechanical properties of porous media [34] during the relative water–gas seepage in pressurised fractured coal rock bodies. The fractured coal rock body is mainly subject to axial pressure and pore fluid pressure, and a schematic diagram of the stress effect on any cross section during seepage is shown in Figure 13.

From Figure 13, the magnitude of the effective stress is closely related to the pore structure, skeletal stress, and pore pressure. Terzaghi [35], in his study of the stress characteristics of aquifers, used the skeletal stress as the effective stress and gave its expression as:(9)$$\sigma_{e}=\sigma -p$$
where: *σ*_e_ is the effective skeleton stress; *σ* denotes the applied stress; and *p* is the hydrostatic pressure.

However, numerous scholars have found that this effective stress fails to respond to changes in the pore structure within the fractured body [36,37]. Therefore, a correction for Terzaghi’s equation is needed in rock percolation scenarios; Biot [38,39] et al. proposed an effective stress equation that considers the effect of the pore structure of the rock body:(10)$$\sigma_{e}=\sigma -\alpha p$$
where α is the Biot coefficient (this is interchangeable with porosity ϕ).

While the above equation is mainly used to calculate the effective stresses carried by the skeleton, Brink and Heymann [40] modified it by treating the skeleton load as a measurable constant value to give an alternative expression for the effective stresses in fractured coal rock:(11)$$\sigma_{e}=\left(1-\phi \right)\sigma +\phi p$$

Considering water and gas as a mixed fluid, the permeability of the water–gas mixture in a fractured coal body satisfies the generalised form of Darcy’s law:(12)$$K=\frac{\mu QH_{s}}{AP}$$
where *K* is the total permeability; *μ* is the viscosity coefficient of the mixed body; *Q* is the total flow rate of the mixed body; and *P* stands for the seepage pressure.

Based on the value of the pore pressure during the seepage test as the magnitude of the seepage pressure, i.e., the association yields an expression for the effective stress and total permeability without regard to seepage pressure as follows:(13)$$\sigma_{e}=(1-\phi )\sigma +\frac{\mu QH_{s}\phi }{A}\cdot \frac{1}{K}$$

From the above equation, the correlation between effective stress and total permeability is obtained. To study the variation in total permeability with effective stress, the relationship between total permeability and effective stress at different seepage pressures is given in Figure 14.

As seen from Figure 14, the total effective permeability decreases as the effective stress increases, and the relationship between the effective stress and the total effective permeability satisfies an exponential function. This shows a strong correlation of the relationship with the effect of the porosity of the sample on the total permeability during the increase in the effective stress: with the increase in the effective stress, the porosity of the sample gradually decreases and the quality of the fluid percolation channel decreases, decreasing the total permeability. The dependence of the total permeability on the effective stress is consistent with the phased evolution of the pore structure of the samples.

In the initial stage, when the effective stress increases, the total effective permeability decreases rapidly. This corresponds to the stage of rapid reduction in sample porosity; during the period when the effective stress increases at a later stage, the total effective permeability decreases less, corresponding to the stable stage of sample porosity. When the sample forms a stable pressure-bearing structure, the pore distribution is stable, and the permeability tends to be stable. The total permeability and effective stress conform to the exponential function relationship, the exponent is constant at −1, and the correlation coefficient exceeds 0.96. This verifies the correctness of the formula between permeability and effective stress, and to simplify the effect of seepage pressure, the normalisation of the fitted formula can be obtained thus: *K* = 177.9 $\sigma_{{}_{e}}^{-1}$ + 213.5.

### 4.6. Evolutionary Characteristics of the Pore Characteristics Parameters of Seepage Systems

The effect of moisture on the permeability characteristics of fractured coal rock bodies can be divided into two categories: on the one hand, the wetting and other effects of moisture on the coal body can affect the pore structure characteristics of the coal body during pressure bearing. On the other hand, the competing adsorption characteristics of moisture and gas can affect the gas adsorption capacity, while the attachment of moisture to the surface of the coal rock body changes the roughness of the surface of the permeability channel, thereby affecting the flow capacity for gas. Further research is required into the effect of water content on the permeability of fractured coal bodies. In this section, the relationship between water content on permeability and non-Darcian flow factors is given for axial loads of 10 kN, 20 kN, and 30 kN.

#### 4.6.1. Effect of Porosity on Permeability

For porous media such as fractured coal samples, porosity is an important parameter in determining the permeability characteristics of the sample, and the variation in the pore structure of the fractured coal rock body is characterised by different characteristics at different water contents. Additionally, this section also selects water content of 0, 5%, and 10% as examples to give the relationship between porosity and permeability at different water contents (Figure 15).

The fitted curves of porosity and permeability also indicate that the effect of water content on permeability is mainly caused by influences on the pore structure of the sample. The effect of porosity on permeability arises mainly because, when the porosity of the sample is large, the pore structure of the sample is looser, the number and penetration of internal seepage channels are better, and the rate of seepage is greater. As the pore space of the sample is gradually compacted, the number and permeability of seepage channels decrease, causing a decrease in the permeability of the sample. According to the fitting results of porosity and permeability, the relationship between the two satisfies, and the correlation coefficient exceeds 0.98. According to the Kozeny–Carman equation [41], it is known that the permeability of porous media is related to the porosity and specific surface of the samples:(14)$$K=\frac{\phi^{3}}{C{\left(1-\phi \right)}^{2}S^{2}}$$
where: *C* and *S* are the *K*-*C* constants and the specific surface area of the solid phase, respectively.

In performing calculations using the above equation, scholars have demonstrated that the KC constant is not a true constant, but is only a constant with respect to porosity, and the results of this study similarly confirm this. The cubic theorem [42] is satisfied between permeability and porosity through analysis of the fitted parameters, as given by:(15)$$K=K_{0}{\left(\frac{\phi }{\phi_{0}}\right)}^{3}$$
where $\phi_{0}$ represents the reference porosity, generally taken as the initial porosity; *K*_0_ is the permeability when the porosity is $\phi_{0}$. The mixed flow of water–gas within the peripore-fractured coal body is shown in Figure 16.

#### 4.6.2. Non−Darcian Flow Pore Permeability Properties

As the porosity and water content increase, the connectivity and number of seepage channels in the fractured coal body increase. The water film effect caused by water adsorption to the surface of the channels leads to a reduction in the rate of seepage resistance to the flow of gas, an increase in the flow of gas, and a weakening of the non−Darcian flow phenomenon in the sample. The effect of porosity on the non−Darcian flow factor of the percolation system of fractured coal media is shown in Figure 17.

The fitted relationship between porosity and non−Darcian flow factor shows that the relationship is exponential, and by analysis of the fitted coefficients, the relationship can be expressed as:(16)$$\beta =\beta_{w0}{\left(\frac{\phi }{\phi_{0}}\right)}^{-3}$$
where $\phi_{0}$ is the reference porosity, generally taken as the initial porosity; *b*_w0_ is the non−Darcian flow factor of the water−bearing fractured coal body at the reference porosity, where *β*_w0_ = *β*_0_ + 6 × 10^8 ^*W*.

For non−Darcian seepage, the effect of porosity on the non−Darcian flow factor also warrants further investigation. The relationship between the non−Darcian flow factor and porosity is given. The non−Darcian flow factor *β* of the fractured coal bodies ranges from 107 to 108 (in terms of order of magnitude), and none of the values of *β* are negative. Meanwhile, the non−Darcian flow factor of the sample decreases with increasing porosity, while the non−Darcian flow factor of the sample decreases with increasing water content, and the effect of water content on *β* only affects the initial stage, and the gradient of the effect of water content on the initial non−Darcian flow factor is 6 × 10^8^.

## 5. Conclusions

(1)At the early stage of extraction, by increasing the negative pressure of extraction, the drainage can be accelerated to reduce the saturation of water in the coal seam, which in turn can effectively increase the relative permeability of gas. The change in gas effective permeability and relative permeability at different porosity levels has an opposite trend, which is mainly due to the competitive permeability between water and gas, and the reduction in gas effective permeability is generally smaller than the reduction in absolute permeability.(2)Under the combined effect of porosity and osmotic pressure, the relative permeability of gas and water shows three stages. The first stage is the gas−dominated stage, in which the gas permeability is significantly greater than the water permeability; the second stage is the isotonic stage, in which the effective permeability and relative permeability of both water and gas are closer and will appear as isotonic values; the third stage is the water−dominated stage, in which the water permeability is significantly greater than the gas permeability.(3)The total effective permeability decreases with increasing effective stress, satisfying *K* = 177.9 + 213.5. The dependence of the total permeability on the effective stress is closely related to the stage in the evolution of the pore structure, with the permeability also showing a rapid decrease during the stage of rapid porosity reduction.(4)Based on the KC equation, the relationship between porosity, permeability and the non−Darcy factor for fractured coal bodies was obtained. The effect of water content on permeability is mainly reflected in the change in test porosity, while water content affects the value of non−Darcy factor at the initial moment and does not affect the evolution pattern of β. The above studies have been carried out so far, and others of interest will be further discovered in future studies.

## Figures and Tables

**Figure 1 ijerph-19-13609-f001:**
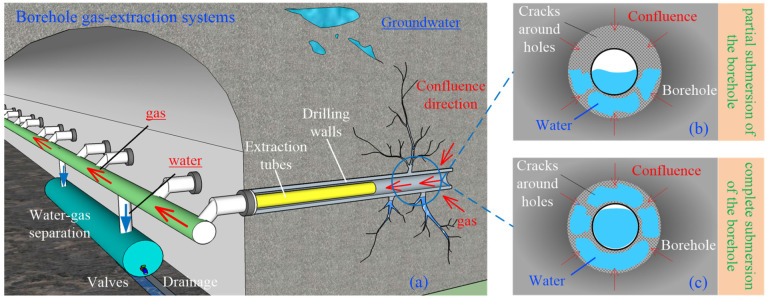
Schematic diagram of the effect of seam water on the permeability of the coal body around a gas extraction borehole. (**a**) Borehole gas extraction systems; (**b**) Partially submerged borehole; (**c**) Fully submerged borehole.

**Figure 2 ijerph-19-13609-f002:**
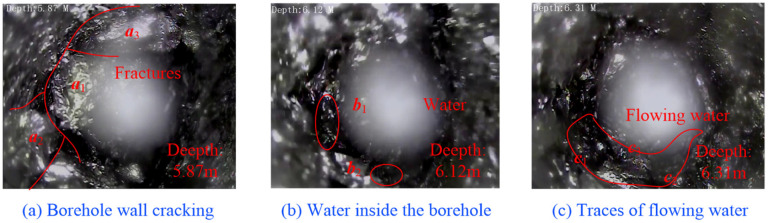
Borehole view of borehole deformation and damage after groundwater erosion.

**Figure 3 ijerph-19-13609-f003:**
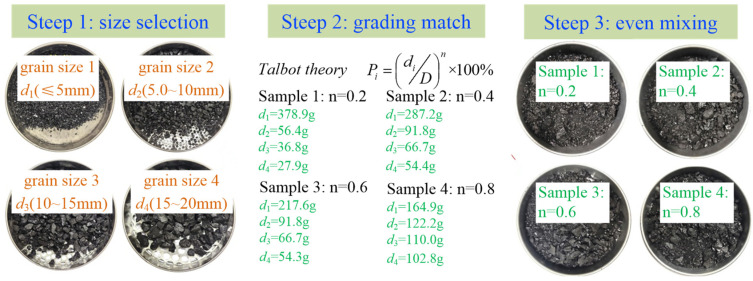
Percentage of particles and preparation process of samples with different grading structures.

**Figure 4 ijerph-19-13609-f004:**
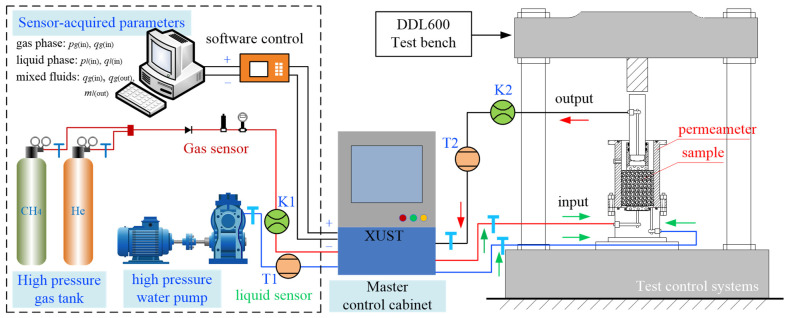
Schematic design for testing water–gas two-phase seepage parameters in fractured coal bodies.

**Figure 5 ijerph-19-13609-f005:**
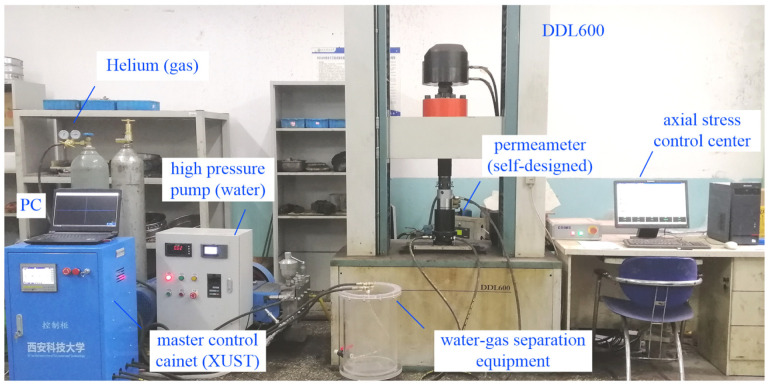
Water–gas two phase seepage test system for fractured coal bodies.

**Figure 6 ijerph-19-13609-f006:**
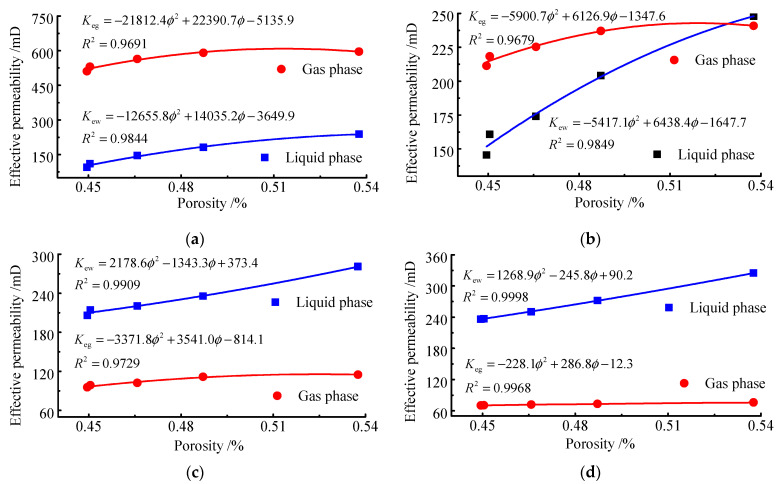
Effect of porosity on effective permeability. (**a**) *p* = 0.2 MPa; (**b**) *p* = 0.4 Mpa; (**c**) *p* = 0.6 Mpa; (**d**) *p* = 0.8 Mpa.

**Figure 7 ijerph-19-13609-f007:**
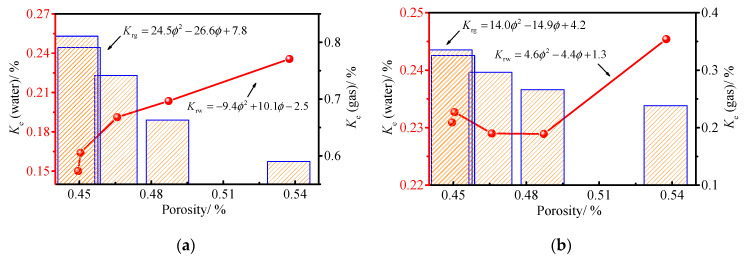
Effect of porosity on relative permeability. (**a**) The seepage pressure is 0.2 MPa; (**b**) the seepage pressure is 0.4 Mpa; (**c**) the seepage pressure is 0.6 Mpa; (**d**) the seepage pressure is 0.8 Mpa.

**Figure 8 ijerph-19-13609-f008:**
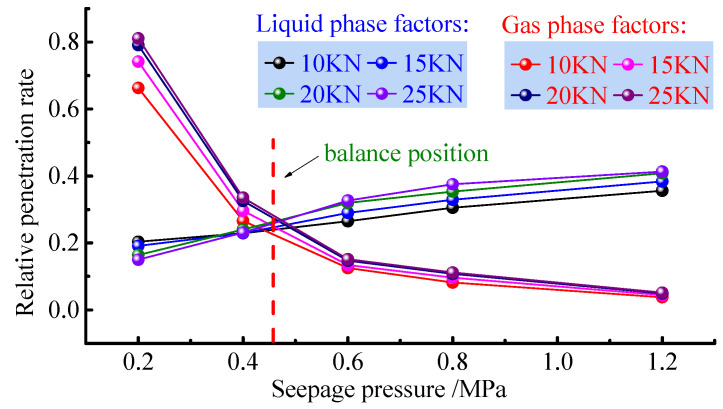
Relationship between relative permeability and seepage pressure.

**Figure 9 ijerph-19-13609-f009:**
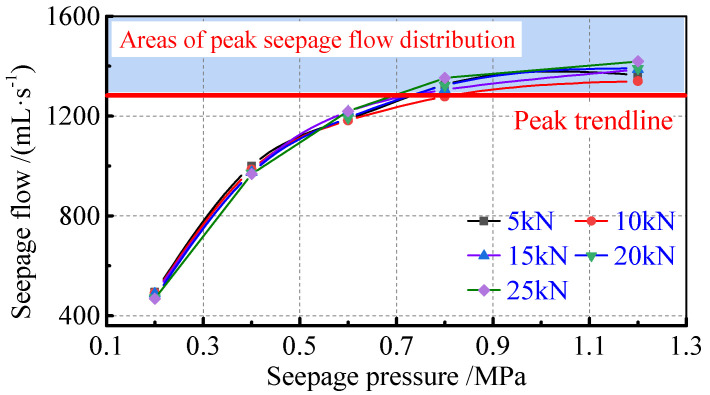
Influence of seepage pressure on gas flow in the seepage system of fractured coal media.

**Figure 10 ijerph-19-13609-f010:**
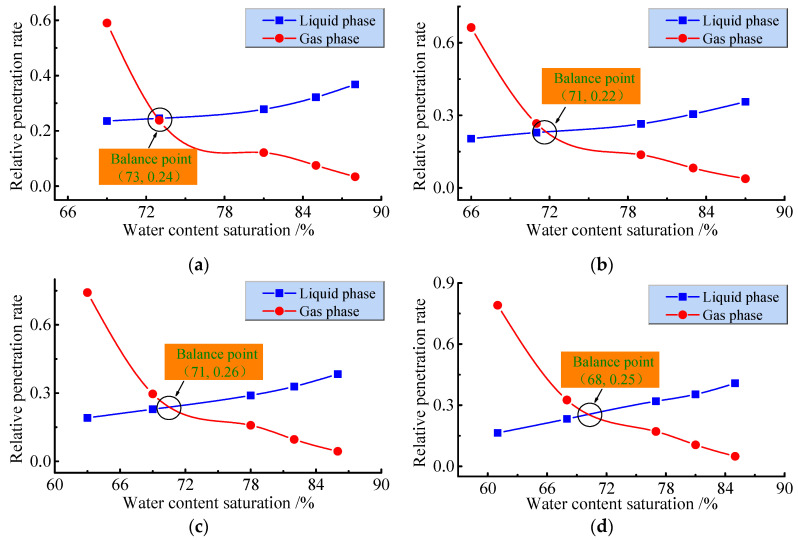
Relationship between relative permeability and water content saturation. (**a**) Sample ZS-01; (**b**) Sample ZS-02; (**c**) Sample ZS-03; (**d**) Sample ZS-04; (**e**) Sample ZS-05.

**Figure 11 ijerph-19-13609-f011:**
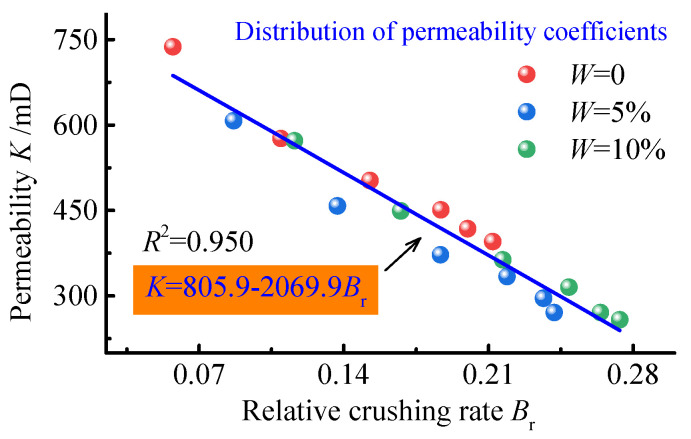
Effect of relative crushing rate on permeability.

**Figure 12 ijerph-19-13609-f012:**
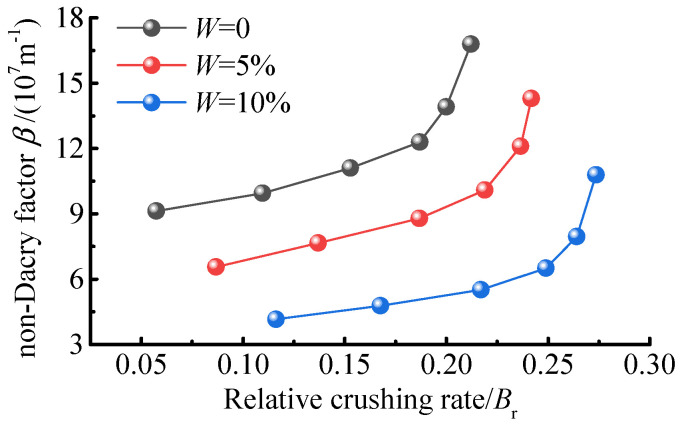
Effect of relative coal crushing rate on *β*.

**Figure 13 ijerph-19-13609-f013:**
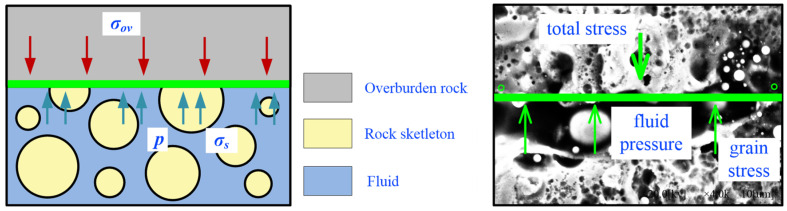
Schematic diagram of the stresses acting on an arbitrary section during seepage.

**Figure 14 ijerph-19-13609-f014:**
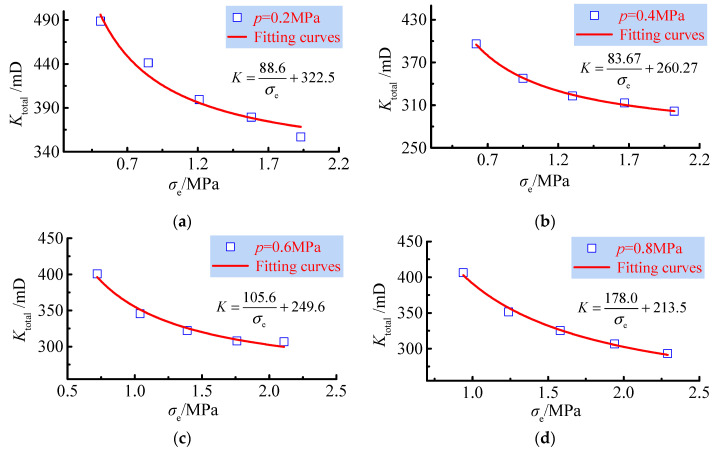
Total permeability versus effective stress. (**a**) The seepage pressure is 0.2 MPa; (**b**) the seepage pressure is 0.4 MPa; (**c**) the seepage pressure is 0.6 MPa; (**d**) the seepage pressure is 0.8 MPa.

**Figure 15 ijerph-19-13609-f015:**
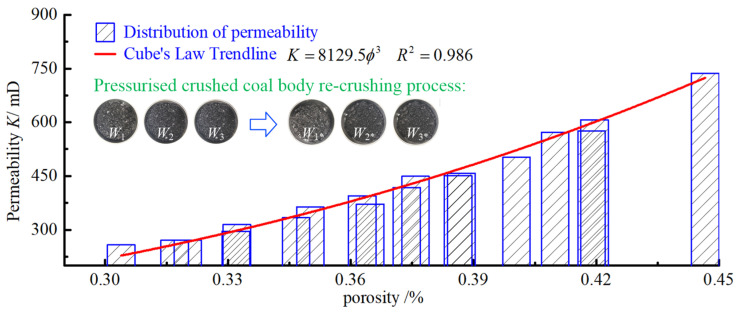
Effect of porosity on permeability.

**Figure 16 ijerph-19-13609-f016:**
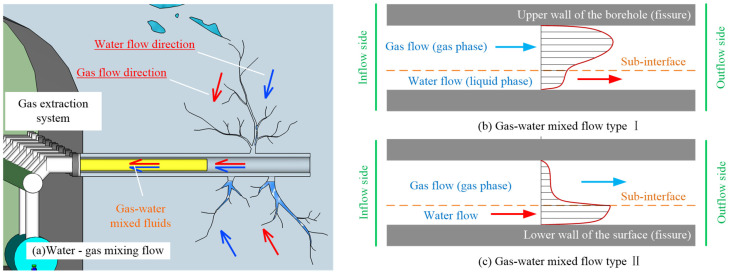
Schematic diagram of water–gas mixing flow within the perihole-fractured coal body.

**Figure 17 ijerph-19-13609-f017:**
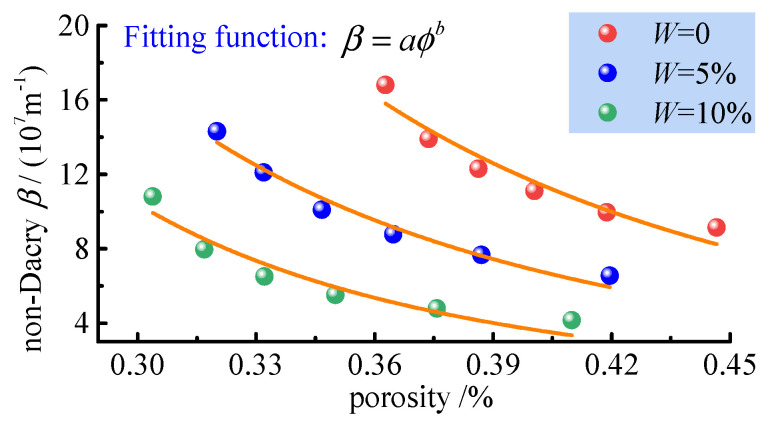
Effect of porosity on the non−Darcian flow factor of the percolation system of fractured coal media.

**Table 1 ijerph-19-13609-t001:** Comparison of relative coal crushing rate and parameters fitted to permeability parameters.

Variable	Fitted Curves	R^2^	Standard Error
W = 0.0%	y = 9.7 × 10^7^ + 4.2 × 10^12^ x^7.1^	0.954	1.791
W = 5.0%	y = 7.2 × 10^7^ + 1.4 × 10^12^ x^7.0^	0.931	2.490
W = 10%	y = 4.7 × 10^7^ + 2.2 × 10^12^ x^15^	0.952	3.447

## Data Availability

All authors approved the publication of the paper. The data used to support the findings of this study are available from the corresponding author upon request.

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
