# Peer review of "Experimental Investigation of the Effect of Groundwater on the Relative Permeability of Coal Bodies around Gas Extraction Boreholes"

_ijerph, 2022, doi:10.3390/ijerph192013609_

Round 1

Reviewer 1 Report

The effect of groundwater on the relative-permeability of coal bodies around gas extraction boreholes was studied in the laboratory. The research results can be used as a reference for coal seam gas drainage around aquifers. However, the experimental fractured coal samples should be prepared according to their seepage pressure to the mine water as the standard, not the particle size distribution of the coal samples. In 3.1.2 Material preparation, the proof of sample preparation is not sufficient in the theoretical and empirical basis. There are also the following problems with this article:

1.The keywords do not meet the requirements. Words with broad meanings are not suitable as keywords.

2.References [1] - [7] are not reasonably cited. The scientific research results of these references should be cited, but they are not.

3.In the introduction, the cited references are insufficient or not fully cited. The previous research results are not fully discussed, the advantages and disadvantages of the previous research results are not analyzed, and the existing problems of the previous research are not summarized.

4.The terminology is inconsistent. For examples: fractured coal body, fractured coal rock boady, crushed coal body, crushed coal rock, crushed coal media, coal crushing media, crushed coal medium, crushed coal grains, brocken coal grains, seepage, permeation, permeability, permeation rate, seepage pressure, osmotic pressure, osmolality , sample, specimen, “water-gas mixing flow, Gas-water mixed fluids.

5.3.1.3 Experimental procedure should be after “3.2.2. Experimental equipment”.

6.In the part of acknowledgment, the authors affirm that this research is supported by two national natural science funds, which is unreasonable. Please select one fund project that is closely related to this research and really supports this research for acknowledgment.

7.English writing needs to be improved. English expression does not conform to English expression habits, and there are errors in grammar and morphology. There are misused words in the text, for example gas pre-pumping should be gas pre-drainage. Some sentences fail to express the meaning and the terms are inconsistent. The English language needs extensive revision.

8.The quality of the figures is poor. The texts in the figures are confused, and the terms in the figures are not consistent with the terms in the titles of the figures.

9.The article structure is not well organized, please reorganize the article structure.

Author Response

Dear Editor:

First, I sincerely thank the reviewers and editors for coming to process my manuscript. Further, the comment letter of expert, and I have responded to each comment point by point as follows.

The effect of groundwater on the relative-permeability of coal bodies around gas extraction boreholes was studied in the laboratory. The research results can be used as a reference for coal seam gas drainage around aquifers. However, the experimental fractured coal samples should be prepared according to their seepage pressure to the mine water as the standard, not the particle size distribution of the coal samples. In 3.1.2 Material preparation, the proof of sample preparation is not sufficient in the theoretical and empirical basis. There are also the following problems with this article:

1.The keywords do not meet the requirements. Words with broad meanings are not suitable as keywords.

Response: Many thanks for this specialist question, the key words I have changed, "Groundwater; Energy extraction; Permeability; Effective stress; Gas seepage" replaced with "Extraction hole; Drilling failure; Permeability; Triaxial stress state; Gas seepage flow process".

2.References [1] - [7] are not reasonably cited. The scientific research results of these references should be cited, but they are not.

Response: The citations in the text are indeed very irregular. By re-reviewing the literature and reading some of the well-established ideas in the relevant studies. The corresponding correct citation format has been added to the text and citations to the literature have been added and revised in the appropriate places in the text.

  1. In the, the cited references are insufficient or not fully cited. The previous research results are not fully discussed, the advantages and disadvantages of the previous research results are not analyzed, and the existing problems of the previous research are not summarized.

Response: Yes, thank you very much for the expert review. The 'introduction' section of the manuscript was not justified. This section has been carefully revised through further literature review and data collection. The conclusions of the experts in the relevant fields have been added, and the research results of the experts have been confirmed. Through such a review, I have revised the "introduction" section. The text has been revised.

  1. The terminology is inconsistent. For examples: “fractured coal body, fractured coal rock boady, crushed coal body, crushed coal rock, crushed coal media, coal crushing media, crushed coal medium”, “crushed coal grains, brocken coal grains”, “seepage, permeation”, “permeability, permeation rate”, “seepage pressure, osmotic pressure, osmolality” , “sample, specimen”, “water-gas mixing flow, Gas-water mixed fluids”.

Response: The use of technical terms in the manuscript is not very standardized. By checking the whole text, the various expressions have been harmonised. The correct expressions are: fractured coal rock boady, seepage, sample, water-gas mixing flow.

  1. “3.1.3 Experimental procedure” should be after “3.2.2. Experimental equipment”.

Response: Thanks to expert advice, I have placed "Experimental procedure" in the manuscript in the designated place.

  1. In the part of acknowledgment, the authors affirm that this research is supported by two national natural science funds, which is unreasonable. Please select one fund project that is closely related to this research and really supports this research for acknowledgment.

Response: Yes, by comparison, the real support for the study is the Shaanxi National Natural Science Foundation project "Fracture evolution and water-gas coupled permeability mechanism of coal bodies around extraction wells" (Grant No. 2021jm-390). 2021JM-390). I have removed other content that is not of the strongest relevance.

  1. English writing needs to be improved. English expression does not conform to English expression habits, and there are errors in grammar and morphology. There are misused words in the text, for example “gas pre-drainage” should be “gas pre-drainage”. Some sentences fail to express the meaning and the terms are inconsistent. The English language needs extensive revision.

Response: Yes, the use of the word in this place is not very standard. I have changed "gas pre-pumping" to "gas pre-drainage".

  1. The quality of the figures is poor. The texts in the figures are confused, and the terms in the figures are not consistent with the terms in the titles of the figures.

Response: By checking the whole text, I have revised the expressions of the technical terms. The most important words are "fractured", "crushed" and "broken". By distinguishing between meanings, I have also corrected any inconsistencies.

  1. The article structure is not well organized, please reorganize the article structure.

Response: By reading the article again, I have revised the inappropriate sections as required. And have thoroughly checked the paper. Many thanks to the experts for their careful annotations.

Reviewer 2 Report

These authors have presented a experiment study on a water-gas two-phase permeation test, in order to determine the relative permeation parameters of the crushed coal media permeation system.  Overall, the topic is important to evaluate the effect of groundwater on the permeability of the coal body around a gas extraction borehole. 

Detailed comments are as followings.

1. Altough it is essential to perform an indepth analysis of the damage and permeation pattern of the coal body around the dip borehole, this work has less discussion directly related to the safety issues of gas extraction from dip  boreholes 

2.The experimental equipment and experiment design described in Figures 4&5 appear differently,especially for the part of permeameter, the layout of the two seems to be opposite.

3. In section 4.1,The result of porosity around 0.51 is missing, which makes it impossible to determine whether the fitting formula is valid, especially the curve with non monotonic change as shown in Figure 7.

4. The formation of curves and legend texts in Figure 6 are not uniform. In section 4.6, the left half of Figure 16 is absolutely comes from part of Figure 1, and it is suggested to degenerate.

Author Response

Dear Editor:

First, I sincerely thank the reviewers and editors for coming to process my manuscript. Further, the comment letter of expert, and I have responded to each comment point by point as follows.

These authors have presented a experiment study on a water-gas two-phase permeation test, in order to determine the relative permeation parameters of the crushed coal media permeation system.  Overall, the topic is important to evaluate the effect of groundwater on the permeability of the coal body around a gas extraction borehole. Detailed comments are as followings.

  1. Altough it is essential to perform an indepth analysis of the damage and permeation pattern of the coal body around the dip borehole, this work has less discussion directly related to the safety issues of gas extraction from dip boreholes.

Response: The commentary mentions thanks to the experts for their suggestions for revision. By checking the "introduction" and "conclusion" of the manuscript, I found that the presentation of the significance of the study was rather rudimentary. Therefore, I have revised these two sections by further condensing the text and adding a statement about the significance of the study as a guide.

2.The experimental equipment and experiment design described in Figures 4&5 appear differently,especially for the part of permeameter, the layout of the two seems to be opposite.

Response: I should make it clear that the structure of Figures 4 & 5 corresponds exactly to each other and there are no layout problems. I can assure you of this. This is because I designed and assembled the instrument, I developed the software and hardware for the data acquisition and recording part, and I carried out the experiments myself. In the permeameter section, the flow direction of both liquid and gas is bottom-up, through the permeameter, and the two paths converge to one. At the outlet we pass water and gas separation, in which the flow parameters of each phase are cleverly determined. In turn, the analysis of the system is carried out.

  1. In section 4.1,The result of porosity around 0.51 is missing, which makes it impossible to determine whether the fitting formula is valid, especially the curve with non monotonic change as shown in Figure 7. 3.

Response: It should be reiterated that five sets of specimens, ZS-01, ZS-02, ZS-03, ZS-04, and ZS-05, were designed for this test, and five levels of axial pressure (5 kN, 10 kN, 15 kN, 20 kN, and 25 kN) were used. For lateral-limit compression experiments, the axial load is the key factor used to control the initial porosity. In this paper, the stress control method is adopted and the five levels of porosity in the manuscript are obtained by testing under these five levels of stress.

The data corresponding to the five points in the curve are all present and there are no missing data. For the fitting of the curves, a quadratic function fit was used, with two monotonic intervals of the function present. The fitted correlation coefficients all reached above 0.9, indicating that the fit in this specimen is valid.

  1. The formation of curves and legend texts in Figure 6 are not uniform. In section 4.6, the left half of Figure 16 is absolutely comes from part of Figure 1, and it is suggested to degenerate.

Response: The legend and curve labelling in Figure 6 (c) and (d) were found to be incorrect after re-screening. The red line should be the relative permeability distribution of the gas phase and the blue line the relative permeability distribution of the liquid phase. It is only in (c) p = 0.6 MPa; (d) p = 0.8 MPa that the positions of the curves for the gas and liquid phases are flipped, distinguishing them from the curves in (a) p = 0.2 MPa; (b) p = 0.4 MPa. This may be related to the shift in the flow dominance of the gas and liquid phases for the high pore pressure case.

It should be noted that in section 4.6 the left half of Figure 16 is taken from a part of Figure 1, but they are viewed from different angles. Figure 1 is intended to show the downhole borehole gas extraction system and the borehole water immersion problems encountered during extraction, and is the context of this study. Figure 16 is intended to reveal the interaction between liquid and gas during the water-gas mixing flow from a mechanistic point of view, which is the highlight of this study and the main research conclusion revealed.

Reviewer 3 Report

Reviewer Comments

Paper title: Experimental investigation of the effect of groundwater on the relative-permeability of coal bodies around gas extraction boreholes

In the present manuscript in order to determine the permeation parameters of the crushed coal body system around the hole, a water-gas two-phase permeation test was designed to determine the relative permeation parameters of the crushed coal media permeation system.

A manuscript has a practical application and also provides important theoretical for the next studies.

The paper can be accepted for publication after providing the corrections mentioned below.

Point 1. The abstract section sounds us conclusion section. The abstract should follow the MDPI style of structured abstracts:

- Background (place the question addressed in a broad context and highlight the purpose of the study);

- Methods (describe briefly the main methods);

- Results (summarize the article's main findings);

- Conclusion (indicate the main conclusions or interpretations).

Point 2. Keywords need to be modified. Please use words not combinations of words or phrases.

Point 3. In the Introduction section, an enhanced literature review is required. For this study, the authors have used only 19 literature sources. It seems insufficient for such type of research. It will be great if the authors show some description in context – Why it is important to conduct this study? Can the expected result be used or implemented within other coal basins? If yes, then how? What limitations?

Point 4. The aim and the tasks must be highlighted at the end of the Introduction section.

Point 5. Subsection 4.1. Figures, Tables and Schemes seem to be incorrectly titled. You should retitle it.

Point 6. The novelty of the paper must be highlighted in the conclusions section.

Point 7. Please consider the suggested research in your paper when enhancing the literature review.

Shapoval, V., Shashenko, O., Hapieiev, S., Khalymendyk, O., & Andrieiev, V. (2020). Stability assessment of the slopes and side-hills with account of the excess pressure in the pore liquid. Mining of Mineral Deposits, 14(1), 91-99. https://doi.org/10.33271/mining14.01.091

Important issue. In this paper has been theoretically proved that for any pore pressure value in the water-saturated mine rock (or soil) their strength will be less than in their water-free state.

Point 8. In general, the presented article leaves a positive impression and, after eliminating these comments and taking into account the recommendations made, it can be recommended for publication in the journal " International Journal of Environmental Research and Public Health".

Author Response

Dear Editor:

First, I sincerely thank the reviewers and editors for coming to process my manuscript. Further, the comment letter of expert, and I have responded to each comment point by point as follows.

In the present manuscript in order to determine the permeation parameters of the crushed coal body system around the hole, a water-gas two-phase permeation test was designed to determine the relative permeation parameters of the crushed coal media permeation system.A manuscript has a practical application and also provides important theoretical for the next studies.The paper can be accepted for publication after providing the corrections mentioned below.

Point 1. The abstract section sounds us conclusion section. The abstract should follow the MDPI style of structured abstracts:

Response: I would like to thank the experts for their valuable advice. By reviewing the requirements of the journal and downloading the relevant published articles from the journal, I identified a structural problem with the 'abstract'. Through study, I have reorganised the abstract according to the sections Background, Methods, Results, Conclusion. The presentation of the abstract was reorganised.

Point 2. Keywords need to be modified. Please use words not combinations of words or phrases.

Response: Thank you very much for this specialised question, I have changed the keywords from "Groundwater; Energy extraction; Permeability; Effective stress; Gas seepage" to "Extraction hole; Drilling failure; Permeability; Triaxial stress state; gas seepage flow process". The keywords include words and phrases that satisfy both requirements and meet the requirements of a conventional scientific paper.

Point 3. In the Introduction section, an enhanced literature review is required. For this study, the authors have used only 19 literature sources. It seems insufficient for such type of research. It will be great if the authors show some description in context – Why it is important to conduct this study? Can the expected result be used or implemented within other coal basins? If yes, then how? What limitations?

Response: Thanks to expert advice, the literature presented in 'Introduction' is relatively sparse and has been re-studied and re-cited. A wide range of relevant research has been drawn on and some citations have been added. The 'Significance of the study' has been added, and the practical implications of the study have been refined in the manuscript. The study of the permeability characteristics of water-bearing boreholes is expected to provide positive guidance for the extraction of gas from water-rich coal seams.

Point 4. The aim and the tasks must be highlighted at the end of the Introduction section.

Response: Thanks for the expert advice, and by reviewing the literature I have realised that there is a problem with the way the introduction is intended to be written. I have reorganised the language. And added some of the latest research findings.

Point 5. Subsection 4.1. Figures, Tables and Schemes seem to be incorrectly titled. You should retitle it.

Response: Many thanks to the experts for their suggestions, the title of section 4.1 was omitted in the wording and has now been revised to its correct form: "Effect of porosity on relative permeability".

Point 6. The novelty of the paper must be highlighted in the conclusions section.

Response: Firstly, when I submitted the manuscript, I also submitted 'highlights' as an attachment, a document that presents the four highlights of the study (attached). Some of the statements in the conclusion may not have brought out the highlights, but I have refined the points by restating the conclusion.(1)Seam water will seize the perimeter coal gas infiltration channel, making it difficult for perimeter gas to flow into the borehole.(2)The infiltration of formation water accelerates the process of creep deformation of the borehole and the extension of cracks in the perimeter of the coal body.(3)Competitive permeability between water and gas around the submerged borehole.(4)The water content affects the value of the non-Dacry factor at the initial moment and does not affect the evolutionary pattern of β.

Point 7. Please consider the suggested research in your paper when enhancing the literature review.

Response: Okay, no problem. I will adopt the findings of this article and I strongly agree with the points submitted in the article.

Point 8. In general, the presented article leaves a positive impression and, after eliminating these comments and taking into account the recommendations made, it can be recommended for publication in the journal " International Journal of Environmental Research and Public Health".

Response: Okay, thank you very much for several sincere comments from the experts and I will revise the paper line by line in accordance with the experts' comments and submit it within the deadline.

Round 2

Reviewer 1 Report

Confusion and misuse of terms still exist in the text and figures. Some key statements are inaccurate. The main problems are as follows:

 1. The keywords still dont meet the requirement. The suggested keywords are: coal gas control; gas pre-extraction; gas extraction borehole; permeability of coal body; effect of groundwater.

2. In the part of Introduction, the wrongly used terms gas pre-pumping still exist at 2 places in the text.

3. In Figure 2, (b) Wet inside the borehole should be (b) Water inside the borehole .

4. In Figure 2, Why does picture (a) , picture (b) , and picture (C) repeat twice?

5. The title of Figure 8 is Relationship between seepage pressure and relative seepage, but the title of the vertical coordinate is Relative penetration rate, which does not agree with the term involved in the title of the Figure.

6. The title of Figure 10 is Water saturation versus relative permeability, but the title of the vertical coordinate is Relative penetration rate, which does not agree with the term involved in the title of the Figure.

7. The title of Figure 11 is Effect of relative fragmentation rate on permeability, but the title of the horizontal coordinate is Relative crushing rate, which does not agree with the term involved in the title of the Figure.

8. The terms relative rate of fragmentation and relative coal crushing rate" are used in confusion.

9. The expressions the relative permeability of gas” and the relative permeability of water are repeated in the text, which are not exact expressions. They should be“the relative permeability of coal body for gas” and “the relative permeability of coal body for water”.

In the part of funding, we acknowledge the provision of the test platformed by the Key Laboratory of Western Mine Exploitation and Hazard Prevention of the Ministry of Education, where upon tests were successfully completed and data were obtained. should be “We are grateful to the Key Laboratory of Western Mine Exploitation and Hazard Prevention of the Ministry of Education for providing the testing platform, where the tests were successfully completed and the data obtained. 

Author Response

Dear Editor:

First, I sincerely thank the reviewers and editors for coming to process my manuscript. Further, the comment letter contains a total of two expert review comments, and I have responded to each comment point by point as follows.

  1. The keywords still don’t meet the requirement. The suggested keywords are: coal gas control; gas pre-extraction; gas extraction borehole; permeability of coal body; effect of groundwater.

Response: I am very grateful for the advice given by the experts, and I strongly agree with the style of keyword selection, and have made changes in the manuscript.

  1. In the part of “Introduction”, the wrongly used terms “gas pre-pumping” still exist at 2 places in the text.

Response: In the 'Introduction' summary, there was a mistake in the key expression 'gas pre-pumping'. This has been carefully checked and corrected in the appropriate place.

  1. In Figure 2, “(b) Wet inside the borehole” should be “(b) Water inside the borehole” .

Response: Yes, thank you very much for the expert review. There is a problem with the labelling in Figure 2, the correct one should be "(b) Water inside the borehole". I have made the correction in the text.

  1. In Figure 2, Why does picture (a) , picture (b) , and picture (C) repeat twice?

Response: By checking the manuscript, there is no problem with Figure 2 in the final version of the manuscript, please verify.

  1. The title of Figure 8 is “Relationship between seepage pressure and relative seepage”, but the title of the vertical coordinate is “Relative penetration rate”, which does not agree with the term involved in the title of the Figure. 5.

Response: I thank the experts for their suggestions. In the manuscript, I have revised "the relationship between seepage pressure and relative seepage" to "the relationship between relative permeability and seepage pressure".

  1. The title of Figure 10 is “Water saturation versus relative permeability”, but the title of the vertical coordinate is “Relative penetration rate”, which does not agree with the term involved in the title of the Figure. 6.

Response: I thank the expert for his suggestion. In the manuscript I have changed "water saturation vs. relative permeability" to "relative permeability vs. water content saturation".

  1. The title of Figure 11 is “Effect of relative fragmentation rate on permeability”, but the title of the horizontal coordinate is “Relative crushing rate”, which does not agree with the term involved in the title of the Figure. 7.

Response: Yes, thanks for the expert advice. In the manuscript, I have changed "Effect of relative fragmentation rate on permeability" to "Effect of relative crushing rate on permeability".

  1. The terms “relative rate of fragmentation” and “relative coal crushing rate" are used in confusion.

Response: By checking the entire text, I have revised the terminology. The most important term, "fragmentation", is correctly "relative coal crushing rate". Checked and corrected the entire text.

  1. The expressions “the relative permeability of gas” and “the relative permeability of water” are repeated in the text, which are not exact expressions. They should be“the relative permeability of coal body for gas” and “the relative permeability of coal body for water”.

Response: By reading the article again, I would like to thank the experts for their comments. The permeability in the manuscript is a calculated permeability, which is related to the physical properties of the fluid medium tested. So the way it is presented in the original article is professional.

In the part of funding, “we acknowledge the provision of the test platformed by the Key Laboratory of Western Mine Exploitation and Hazard Prevention of the Ministry of Education, where upon tests were successfully completed and data were obtained.” should be “We are grateful to the Key Laboratory of Western Mine Exploitation and Hazard Prevention of the Ministry of Education for providing the testing platform, where the tests were successfully completed and the data obtained. ”

Response: Thank you for the expert advice. I have made the changes.

Reviewer 3 Report

Dear authors,

I am more than satisfied with the corrections provided by you.

This study is an important contribution to the research field.

Congratulations to the authors.

Author Response

Dear Editor:

First, I sincerely thank the reviewers and editors for coming to process my manuscript. Further, the comment letter contains a total of two expert review comments, and I have responded to each comment point by point as follows.

Reviewer 3

Dear authors,

I am more than satisfied with the corrections provided by you.This study is an important contribution to the research field.Congratulations to the authors.

Response: I would like to thank the experts for their careful comments on my manuscript, and the quality of the paper has improved under their guidance.
